# Removal of Antibiotic Resistance Genes, Class 1 Integrase Gene and *Escherichia coli* Indicator Gene in a Microalgae-Based Wastewater Treatment System

**DOI:** 10.3390/antibiotics11111531

**Published:** 2022-11-02

**Authors:** Abdullahi B. Inuwa, Qaisar Mahmood, Jamshed Iqbal, Emilie Widemann, Sarfraz Shafiq, Muhammad Irshad, Usman Irshad, Akhtar Iqbal, Farhan Hafeez, Rashid Nazir

**Affiliations:** 1Department of Environmental Sciences, COMSATS University Islamabad (CUI), Abbottabad Campus, Abbottabad 22060, Pakistan; 2Department of Microbiology, Faculty of Life Sciences, College of Natural and Pharmaceutical Sciences, Bayero University Kano, Kano 700006, Nigeria; 3Department of Biology, College of Science, University of Bahrain, Sakhir P.O. Box 32038, Bahrain; 4Centre for Advanced Drug Research, COMSATS University Islamabad (CUI), Abbottabad Campus, Abbottabad 22060, Pakistan; 5Department of Pharmacy, COMSATS University Islamabad (CUI), Abbottabad Campus, Abbottabad 22060, Pakistan; 6Institut de Biologie Moléculaire des Plantes, CNRS-Université de Strasbourg, 67084 Strasbourg, France; 7Department of Anatomy and Cell Biology, University of Western Ontario, 1151 Richmond St., London, ON N6A5B8, Canada

**Keywords:** microalgae, municipal wastewater, antibiotic resistance genes, *Escherichia coli*

## Abstract

Microalgae-based wastewater treatment systems (AWWTS) have recently shown promise in the mitigation of antibiotic resistance genes (ARGs) from municipal wastewater (MWW). However, due to the large number of ARGs that exist in MWW, the use of indirect conventional water quality parameters to monitor ARGs reduction in wastewater would make the process less burdensome and economically affordable. In order to establish a robust relationship between the ARGs and water quality parameters, the current study employed different microalgae strains in monoculture (CM2, KL10) and multi-species combinations (CK and WW) for the MWW treatment under outdoor environmental conditions. The studied genes were quantified in the MWW influents and effluents using real-time PCR. All the cultures substantially improved the physicochemical qualities of the MWW. Out of the 14 genes analyzed in this study, *tetO*, *tetW*, *tetX* and *ermB* were decreased beyond detection within the first 4 days of treatment in all the cultures. Other genes, including *bla*_CTX_, *sul1*, *cmlA*, *aadA*, *int1* and *uidA* were also decreased beyond a 2 log reduction value (LRV). The mobile genetic element, *int1*, correlated positively with most of the ARGs, especially *sul1* (r ≤ 0.99, *p* < 0.01) and *aadA* (r ≤ 0.97, *p* < 0.01). Similarly, the *Escherichia coli* indicator gene, *uidA,* correlated positively with the studied genes, especially with *aadA*, *bla*_CTX_, *bla*_TEM_ and *cmlA* (r ≤ 0.99 for each, *p* < 0.01). Some of the studied genes also correlated positively with total dissolved solids (TDS) (r ≤ 0.98, *p* < 0.01), and/or negatively with total suspended solids (TSS) (r ≤ −0.98, *p* < 0.01) and pH (r ≤ −0.98, *p* < 0.01). Among the tested cultures, both monocultures, i.e., KL10 and CM2 were found to be more consistent in gene suppression than their multi-species counterparts. The findings revealed water quality parameters such as TDS, TSS and *E. coli* as reliable proxies for ARGs mitigation in AWWTS and further highlight the superiority of monocultures over multi-species cultures in terms of gene suppression from the MWW stream.

## 1. Introduction

Antibiotic resistance genes (ARGs) are one type of the emerging contaminant which pose serious public health issues globally. According to the United Nations (2019), “Antimicrobial resistance is one of the most urgent health risks of our time and threatens to undo a century of medical progress”. This emerging threat imposed by antibiotic resistance has earned it a mention in the list of emerging contaminants or contaminants of emerging concern [1,2,3]. The high propensity for the amplification and evolution of ARGs in urban wastewater (WW) earns this environment the slogan of a genetic reactor [4]. In a global surveillance of urban sewage across six countries, it was found that ARGs diversity varied from region to region and country to country. However, ARGs encoding resistance toward macrolides, tetracycline, aminoglycosides, beta-lactams and sulfonamides were the most abundant of all. Previous studies have also reported the possible enrichment of ARGs in conventional WW treatment (WWT) plants [5,6]. For instance, chlorine disinfection was reported to raise the levels of *ermB, tetA, tetB, tetC, sul1, sul2, sul3, ampC, aph(2’)-Id), katG* and *vanA* [7]. Likewise, it was reported that as a biological treatment step in many WW treatment facilities, activated sludge enriches ARGs most probably because it supports the multiplication of bacteria thereby increasing the chances of horizontal gene transfer [8,9]. This potential for gene-enrichment has made ARGs an emerging threat in conventional WWT plants.

Recent studies have preliminarily reported the potential of microalgae-based wastewater treatment system to remove ARGs from MWW [10,11]. This adds to the reputation of the system as a low-cost, green alternative to WWT systems for the removal of nutrients, organic matter, fecal coliforms, fecal streptococci and viruses [12,13,14,15,16,17,18]. However, since routine ARGs monitoring in WW may not be feasible, establishing a nexus between ARGs and wastewater quality parameters would provide a basis for the possible use of wastewater quality indicators as reliable proxies for ARGs load assessment in AWWT systems. 

Therefore, the principal goal of the current study was to investigate the possible relationship of antibiotic resistance genes with bacterial and water quality indicators in a microalgae-based wastewater treatment system.

## 2. Results

### 2.1. Time Course Changes of Water Quality Parameters in Microalgal Cultures

#### 2.1.1. Total Dissolved Solids

The TDS content of the cultures presented a uniform pattern of initial rise at day 4 and then a steady decline afterwards (Figure 1A). The period between days 4 to day 12 witnessed the major TDS reduction in all the cultures. Afterwards, the TDS level in all the cultures remained stationary with some occasional slight fluctuations until day 20. At day 20, the highest TDS reduction (32.25%) was yielded by KL10 culture. However, while the TDS reduction in KL10 differed significantly from that of CM2 (*p* < 0.05), it was still comparable to multi-species cultures (*p* > 0.05). This implies that while the TDS reduction of the monocultures differed, co-cultures remained comparable regardless of their diversity and/or origin.

#### 2.1.2. Electric Conductivity

The EC of the cultures presented an initial rise from 1229 ± 1 at day 0 up to 1343.33 ± 11.6 μS/m in CM2 culture at day 4 (Figure 1B). This was followed by a downward trend, which persisted until day 12. Afterwards, the trend remained somewhat stationary until the end. A visible exception was KL10, which showed some undulating patterns from day 8 to day 16 before finally becoming stationary. The trend in EC reduction as well as the reduction performance among the cultures was very much similar to that of TDS.

#### 2.1.3. Total Suspended Solids

The TSS content of all the cultures followed a similar pattern, maintaining a steady rise over the course of the treatment time (Figure 1C). The sharpest rise was between day 0 and day 4 indicating that the cultures progressed to the logarithmic phase instantly with no obvious lag phase. By the 16th day of treatment, the TSS ranged from 1276.7 ± 70.95 to 1630 ± 55.68 mg/L. Afterwards, the TSS accumulation levelled off, and there was no significant difference between the TSS at day 16 and day 20 for all the cultures (*p* > 0.05). The two monocultures occupied the two extremes of the TSS content of all the cultures; although the best of the two (KL10) was still not better than the two multi-species cultures (*p* > 0.05). Conversely, the TSS content of the two multi-species cultures fell within the same range (*p* > 0.05). Overall, the results revealed that no differences in TSS accumulation between the best monoculture (KL10), the synthetic binary consortium and the wild consortium.

#### 2.1.4. pH

The pH of the cultures showed a uniform trend throughout the treatment time (Figure 1D). From day 0 of treatment, the pH of the cultures maintained an upward trend until attaining the peak at day 12. The range in pH for the cultures at that stage was 10.61 ± 0.1 (CM2) to 10.84 ± 0.07 (WW). This change was from the initial weak alkaline pH of 7.71 (day 0), indicating that the growth and activities of the microalgae could have favored the rise in pH of the cultures. Beyond day 12 the cultures witnessed a slight, yet significant, (*p* < 0.05) downward shift in the pH, indicating a possible slowdown in the algal growth. At day 20, the highest pH was observed in CK and KL10 cultures (*p* > 0.05). In coherence with this, the two cultures with least pH also did not differ significantly (*p* > 0.05) and were also of mono and multi-species combination. Together, this indicate that the species diversity of the culture had no noticeable influence on pH change in the cultures.

#### 2.1.5. Temperature

The temperature of the cultures rose from 15.93 °C ± 0.058 to 23.33 °C ± 0.351 in CK culture at day 4 (Figure 1E). Generally, the temperature of all the cultures peaked at day 4 of incubation, possibly indicating vigorous metabolic activities. Subsequently, the temperatures dropped at day 8 and then rose steadily until day 16. Afterwards, there was a sharp drop, leading to all-time low temperatures for all the cultures at day 20 (12.53 °C ± 0.153 in KL10). Evidently, from day 0, the temperature of all the cultures remained within the mesophilic range until day 16, which coincided with the period of more active algal growth. This suggests that the active algal growth might be responsible for the sustenance of mesophilic temperature in the cultures. Comparison of the temperature of the cultures at day 20 revealed no significant differences (*p* > 0.05) between the monoculture with the highest temperature level (CM2) and its multi-species counterpart (CK). 

#### 2.1.6. Nitrate Nitrogen

Initially (day 4), the NO_3_-N level in all the cultures rose sharply (in the range of 94.0 to 94.41% rise) (Figure 1F). However, these elevated NO_3_-N levels could not last beyond day 8 of the treatment, which witnessed a sharp decline in the NO_3_-N levels in all the cultures (92.6 to 99.4% reduction). Though fluctuating, this low NO_3_-N level persisted in all the cultures until the end of treatment where the highest (*p* < 0.05) NO_3_-N reduction (82%) was yielded by KL10 culture. These fluctuations in the NO_3_-N levels point at the potentially high activities of nitrifying bacteria in all the cultures. However, simultaneous analysis of nitrifying bacteria and/or their functional genes in the system would shed more light on this.

### 2.2. Occurrence and Relative Abundance of the Genes in the Influent

The PCR assay targeted and detected a total of 11 ARGs (*aadA*, *ermB*, *bla*_CTX,_ *bla*_TEM_, *cmlA*, *floR*, *sul1*, *sul2*, *tetO*, *tetW* and *tetX*), 16S rRNA, *int1* gene and *uidA* (*Escherichia coli* gene) in the influent. Over the course of the microalgal cultivation in the MWW, these genes were quantified both in influents and effluents using RT-PCR. All the target genes were detected in the influent, albeit at varied levels ranging from 1.63 × 10^−5^ (*ermB*) to 2.476 × 10^−2^ (*int1*), indicating the importance of MWW as reservoirs of ARGs pollution (Figure 2). The high level of *int1* detected in the influent (up to 49% of the total genes’ relative abundance) suggests not only the high likelihood for horizontal gene transfer but also the importance of *int1* as an indicator of overall ARGs pollution in MWW. Narrowing down to ARGs, *sul1* had highest relative abundance of all the ARGs detected in the influent and accounted for 31.4% of the total relative abundance of the studied genes. This was followed by *bla*_CTX_ and *bla*_TEM_, which collectively accounted for 9.2% of the total relative abundances of the studied genes. Notably, tetracycline resistance genes had one of the least relative abundance of all the genes. This pattern of relative abundance of the ARGs in influent partly agrees with a previous report [19] on the high level of resistance to sulfonamides and β-lactams, while disagreeing with the same on the levels of tetracycline resistance.

### 2.3. The Effect of Treatment on the Relative Abundance of Individual Genes

The 20-day-old microalgae treatment of the wastewater resulted in varied changes in the ARGs and indicator genes burden of the wastewater. Visibly, these differences arose from the influences of the individual cultures used, the target gene itself as well as the length of the holding time. By day 4 of the treatment, *ermB, tetO, tetW* and *tetX* (Figure 3) all reduced beyond detection limit of the assay, indicating the efficiency of the system in eliminating the ARGs. Similarly, *bla*_CTX_, *bla*_TEM_, *cmlA, floR* (Figure 4) *and uidA* (Figure 5B) were all reduced significantly (*p* > 0.05) in all the cultures over the same time period. Conversely, sulfonamide resistance genes showed persistence in some cultures, only reducing significantly in CM2 and CK (*sul1*) as well as in CK and KL10 (*sul2*) (Figure 4). Finally, the fate of *int1* in the cultures ranged from significant reduction (KL10, *p* > 0.05), insignificant reduction (CM2 and WW, *p* < 0.05) and enrichment (CK). By day 8 of treatment, all the cultures recorded a significant reduction in all the study genes (*p* < 0.05), signifying the efficiency of the system to maintain the performance achieved previously. An exception was observed with *floR* reduction, whose relative abundances in KL10 and WW culture rose above that of the influent (*p* > 0.05). Beyond day 8 of treatment, the reduction of the relative abundances of the genes in all the cultures either improved substantially (*p* < 0.05) or remained unchanged (*p* > 0.05) until the end. Higher relative abundances (*p* < 0.05) of *bla*_CTX_, *bla*_TEM_, *sul2*, *cmlA* and *uidA* were observed in the two multi-species cultures relative to the two monocultures, which revealed comparable (*p* > 0.05) reduction of the same genes. Similarly, the best (*p* < 0.05) final reduction of relative abundances of *sul1* and *aadA* were yielded by CM2, whereas the reduction of *floR* and *int1* (Figure 5) remained comparable (*p* > 0.05) in all the cultures. Overall, the cultures ensured a reduction in the relative abundances of the study genes from MWW; however, with certain genes, the monocultures edged out their multi-species counterparts in terms of performance.

### 2.4. Comparison of the Efficiencies of Holding Times in Reduction of Relative Abundances of Total ARGs, Total int1 and Total uidA

The total ARGs, *int1* and *uidA* relative abundance for each holding time was calculated by summing up the average ARGs abundances of the individual microalgae cultures. The totals of ARGs, *int1* and *uidA* computation were meant to provide a rough overall idea of the fate of the genes in relation to the treatment time. For each gene, the relative abundances yielded by individual replications were summed up to reach at the total.

The total ARGs relative abundances of the cultures maintained a steep, significant successive reductions (*p* < 0.01) from day 0 up to day 8 (Figure 6A). The reduction efficiency slowed down between day 8 and 12, though still significant (*p* < 0.05). Further treatment resulted in insignificant change (*p* < 0.05) in the relative abundance of the gene, implying that treatment beyond day 12 would suffice for appreciable total ARGs removal from the system.

In a slight contrast to the total ARGs reduction, the changes in total *int1* level was characterized by initial enrichment (*p* > 0.05) at day 4 from day 0 (Figure 6B). This was followed by a sharp drop at day 8 (*p* < 0.05). The reduction trend was sustained successively (*p* < 0.05) till day 16. Finally, as observed with total ARGs, treatment until day 20 resulted in insignificant enrichment (*p* > 0.05) relative to the preceding holding time.

Compared with both total ARGs and *int1* gene, the total *uidA* reduction potentials of the system was the most efficient (Figure 6C). As early as day 4 of treatment, there was a drastic drop (*p* < 0.01) in relative abundance of total *uidA* genes from the initial level. The relative abundance of the gene continued to remain at comparably (*p* > 0.05) low level across the holding times until the end of treatment, signifying that treatment beyond four days was unnecessary for the substantial removal of the gene from the system.

### 2.5. Relationships between ARGs, Indicator Genes and Water Quality Parameters

In order to elucidate the possible linkage between the study variables, principal component analysis (PCA) of the variables was carried out. Separate PCA biplots were plotted for the individual cultures, explaining the variation in relative abundance of the study genes as well as the water quality parameters relative to the holding times (Figure 7). The PC1 component of the biplot explains 63.29 to 68.91% of the total variations in the dataset. In all the biplots, the studied genes (except for *int1* in KL10 culture) were clustered together in this component, showing their influence on the total variations explained by the biplot. Furthermore, the vectors of majority of the genes in the component were skewed towards day 0 (the influent), signifying the higher relative abundance of the genes in the influent. Generally, the spatial distribution of the genes in the biplots shows low dissimilarity, resulting in clustering of the genes in the biplots; however, the extent of the clustering was more expressed and consistent with *bla*_CTX_; *bla*_TEM_, *cmlA* and *uidA*. This was further buttressed by significant positive correlations of the genes (r = 0.929 to 0.999, *p* < 0.01) in the monocultures (Appendix A). A similar pattern was observed with the multi-species cultures (r = 0.923 to 0.999, *p* < 0.01), except for the addition of *aadA* to the cluster (Figure 7).

Conversely, the PC2 components of the two monocultures (Figure 7A,B) accounted for 19.04 to 26.11% of the total variations. For all the cultures, the physicochemical water quality parameters were grouped in this component. In addition, *int1* was also grouped in this component (except for KL10). TDS, EC, pH and TSS were the most influencing of the physicochemical parameters. Among these, TDS and EC correlated positively with all the study genes except for *floR* in KL10 and multi-species cultures (Figure 7C,D). Notably, TDS revealed significant positive correlation with *int1* in all the cultures (r = 0.836 to 0.993, *p* < 0.05 or *p* < 0.01). In addition, the correlation of TDS with *sul1* in the two multi-species cultures (r = 0.918 to 0.976, *p* < 0.01) and KL10 culture (r = 0.938, *p* < 0.05) was also significant. 

As opposed to TDS and EC, pH and TSS correlated negatively with many of the studied ARGs. Three genes *sul1*, *int1* and *uidA* (r = −0.829 to −0.982, *p* < 0.05), significantly correlated with pH in the two monocultures, whereas in the multi-species cultures (Appendix A), pH correlated most consistently with *aadA, bla*_CTX_ and *sul1* (r = −0.812 to −0.848, *p* < 0.05). Compared to pH, the negative correlation of TSS with the study genes was relatively more consistent. Notably, in the monocultures *bla*_TEM_, *sul1*, *sul2*, and *uidA* (r = −0.866 to −0.949, *p <* 0.01 or *p <* 0.05) revealed the strongest correlation with TSS. Similarly, in the multi-species cultures, *aadA*, *bla*_CTX_, *bla*_TEM_ and *cmlA* correlated strongly with TSS (r = −0.868 to −0.956, *p* < 0.01 or *p* < 0.05).

It is noteworthy that as one of the two indicator genes analyzed in the current study, *uidA,* showed significant positive correlation (*p* < 0.05) with all the study ARGs (except *floR*) in all the cultures. Aside from its strong positive correlation with *bla*_CTX_, *bla*_TEM_, *cmlA* and *aadA*, as already highlighted, the indicator gene also revealed consistently significant positive correlation with *sul1* (r = 0.813 to 0.983, *p* < 0.05) in the monocultures as well as with *sul2* in CK culture (r = 0.928, *p* < 0.01). The second indicator gene, *int1*, also correlated positively with all the study genes, albeit at varying degrees. The most solid correlation of the gene was observed with *aadA* and *sul1* (r = 0.97 to 0.99, *p* < 0.01) in the monocultures and with sul1(r = 0.95, *p* < 0.01) in WW cultures.

## 3. Discussion

### 3.1. The Effect of Microalgal Cultivation on Water Quality Indicators

TSS accumulation provides a direct way of assessing microalgae growth in a medium [20,21]. Additionally, in the current study, the sharp rise in TSS content of the cultures from day 0 to day 4 highlights the instant adaptation of the microalgae to the wastewater medium and also provides a testimony that the MWW contained necessary nutrients and organic matter necessary for microalgal growth. In the same vein, the TDS content of the influent was already below the National Environmental Quality Standards (NEQS) permissible limit for discharge [22] and hence no need for further digression. However, the final pH value of all the effluents stood above the 6–9 NEQS approved range for MWW effluents [22], hence there may be the need for prior pH adjustment. The final temperature of the cultures did not differ significantly from one another, suggesting that temperature could not have played any significant role in any perceived differences in the performances of the cultures. However, previous studies have reported the temperature of the microalgal culture medium to be a key player in the overall performance of the system [23,24]. 

### 3.2. Occurrence and Abundance of ARGs in the MWW 

The current study tracked 11 different ARGs in an MWW stream, which served as a growth medium for microalgae. The profile of the detected ARGs resemble that of urban sewages, recreational waters and other water bodies of urban origin [2,19,25]. The relative abundance of the genes in the influent varied from the order of 10^−5^ to 10^−2^ and this range was close to what was reported earlier [11]. The class 1 integrase gene, *int1*, was the most abundant of all the genes in this study. High level of this gene was earlier reported in MWWs [11] and estuaries, reflecting its status as a global indicator of ARGs dissemination in wastewaters and other aquatic habitats [2,25,26]. Narrowing down to ARGs, *sul1* had the highest relative abundance of all the genes. The high abundance of *sul1* encountered in this study was consistent with previous studies [11,27]. The two β-lactam resistance genes, *bla*_CTX_ and *bla*_TEM_, were the next most abundant genes detected in this study (10^−3^ order of magnitude). The relative abundance of the two genes was higher than the magnitude reported previously [28]. Alongside *sul1, bla*_CTX_ and *bla*_TEM_ were reported to dominate the ARGs pool of hospital waste-contaminated waters [27,28]. Considering that our sampling point represented a section of the sewer that just passed through Ayub Teaching Hospital, we attributed the high load of *sul1*, *bla*_CTX_ and *bla*_TEM_ to possible contamination by medical waste. On the other extreme end of the order of ARGs detected in this study, tetracycline resistance presented the least relative abundance (only higher than *ermB*). Previous reports have shown that these genes dominate in livestock wastewaters [29]. The low level of these genes encountered herein might therefore be a reflection of typical urban WW in our case dominated by human rather than veterinary wastes. 

### 3.3. The Effect of Treatment on ARGs Relative Abundance

The 20-day old AWWT resulted in substantial decline in the relative abundance of the ARGs, *int1* and *uidA* across various holding times. It was also observed that although the performances of the cultures tend to be comparable in most cases, instances existed where certain cultures outperformed others. The comprehensive reduction pattern based on the classes of the ARGs is given as follows. 

#### 3.3.1. β-lactam Resistance Genes: *bla*_CTX_ and *bla*_TEM_


These genes accord a global presence due to the widespread use of the β-lactam antibiotics [19,30]. In the current study, the genes suffered early reduction in the system with no obvious sign of enrichment along the treatment line. Obviously, the genes showed no sign of persistence in the system, with the best reductions observed at day 20. The LRV of 2.485 observed with *bla*_CTX_ represents a remarkable improvement over slight reduction to enrichment reported in the effluents of urban MWW plants from seven different Croatian cities [30]. It also surpassed 1.27 order of magnitude reduction reported from four different multi-step urban wastewater treatment plants from Harbin, China [31]. As for *bla*_TEM_, the highest LRV of the gene was 1.869. This moderate reduction was also reported in a comparative study where the relative abundance of the gene was lower in algal treatment plant effluent compared with both primary and secondary (activated sludge) treatment systems operating in parallel [23]. However, contrary to this, increased relative abundance of the gene was reported in the effluents of full-scale wastewater treatment plants [28].

#### 3.3.2. Sulfonamides Resistance Genes: *sul1* and *sul2*

Sulfonamides resistance genes tend to persist in the environment and are thought to pre-date the sulfonamide drugs themselves [25]. They have been reported to be among the most dominant ARGs in global wastewaters [19]. In this study, poor reduction of *sul1* gene was observed at day 4, implying that 4-day treatment was not sufficient. Similar observations were reported for both conventional wastewater treatment plants [2,32] and microalgae-based type [11], affirming the persistent attribute of this ARG. The reduction of the gene attained its highest peak (2.35 LRV) at day 16 in the KL10 culture. This magnitude of LRV was higher than the 0.60–1.63 LRV achieved by three conventional wastewater treatment plants [32]. However, at the end of the treatment, the highest reduction dropped to 1.16 LRV, signifying increased relative abundance of the gene. This observation was also the same for all the cultures except CM2. Interestingly, this observation also coincided with a significant drop (*p* < 0.05) in NO_3_-N level from day 16 to day 20 in all the cultures except CM2. This suggests that the rise in the relative abundance of the gene in those cultures and at that particular time might be related to NO_3_-N biological transformation in the system. A similar observation was reported previously in a fabricated system treating aquaculture influent and it was thus hypothesized that the shift in NO_3_-N-related bacterial communities influenced the variations in the relative abundance of the gene [33]. Regardless, at the end of the treatment, the CM2 culture recorded the highest (*p* < 0.05) reduction of the gene (1.45 LRV) than all other cultures, including KL10. This magnitude of reduction was still higher than the 1.14 LRV achieved by artificially-lighted AWWT [11] and increased relative abundance observed in an acidophilic microalga-based wastewater treatment system [23].

As opposed to *sul1, sul2* exhibited less initial persistence and most of the treatments yielded positive reductions by day 4. The highest LRV of the gene at day 20 was 1.4, which was higher than <0.5 LRV reported for the effluents of fine screen, grit chamber, sedimentation tank and UV disinfection steps of four urban wastewater treatment plants but comparable to the 1.2 LRV reported for the cyclic activated sludge system of the same plants [31]. Furthermore, the magnitude of the reduction of the gene represents an improvement relative to slight enrichment observed in aerated lagoon and BNR plants [2] and anoxic-oxic plants [34].

#### 3.3.3. Phenicol Resistance Genes: *cmlA* and *floR*

The phenicol resistance genes were also reported to dominate the ARGs pool of human sewage from cities [19]. The two phenicol resistance genes, *cmlA* and *floR*, are known to encode efflux pumps [1] and exhibit persistent trends in aquatic systems [35]. The two related genes experienced contrasting fates in the system: *cmlA* suffered one of the best ARGs reduction in the system (up to 2.81 LRV), whereas *floR* was the most persistent of all the genes studied herein (−0.461 to 1.07 LRV). Although moderate, the respective LRV of the two genes compared with what was reported (−0.55 to 1.51 for *cmlA* and −0.27 to 0.72 for *floR*) from two drinking water treatment plants, each involving multiple treatment steps [36].

Compared with the subsequent holding times, the removal of *floR* across all the cultures was more pronounced at day 4 of treatment. This implies that the extended treatment might favor the growth and multiplication of the *floR*-host bacteria in the system. The gene has been observed to persist in an electrochemical anaerobic chloramphenicol wastewater treatment unit, a phenomenon attributed to the persistence of the host bacteria in the system [1]. It is therefore possible that *floR*-harboring bacteria possess attributes that enable them to thrive in biological wastewater treatment systems.

#### 3.3.4. Aminoglycosides Resistance Gene: *aadA*

The level of *aadA* gene detected in the current study was similar to that from Swedish wastewaters [11]. Past studies have reported this gene to persist in microalgae-based wastewater treatment plants [11], conventional wastewater treatment plants [8] and manure samples [37,38]. On the contrary, the results herein showed that 16-day treatment was enough to yield appreciable reduction (1.87 to 2.56 LRV) of the ARG in all the cultures. Furthermore, the magnitude of the reduction in the relative abundance of the gene observed herein (3.1 LRV) was the highest for any ARG in the current study.

#### 3.3.5. Tetracycline Resistance Genes: *tetO*, *tetW* and *tetX*

These ARGs have a long-standing history of global presence [19,39]. Interestingly, all the three tetracycline resistance genes analyzed in this study went undetectable by day 4 of treatment. Although the genes had one of the least abundance (in the order of 10^−4^) in the influent, the fact that another gene*-floR* with relative abundance of a similar order persisted in the system proves that other forces beside low relative abundance of the genes in the influent must have contributed to the complete elimination. Furthermore, compared with previous studies, the relative abundance of the genes in the influent was found to be either comparable or 1 order of magnitude below what exists in some previous reports on wastewaters [11,39]. Substantial reduction of *tetW* was also reported in microalgal systems and activated sludge, a phenomenon attributed to the possible low horizontal gene transfer frequency of the gene [11,23]. Similarly, up to 4.6 LRV of *tetO* from septic tank effluent was achieved by flow lateral sand filters [40]. Likewise, up to 4.73 LRV of *tetX* was achieved in a membrane bioreactor treating MWW [41]. In the current study, the possible explanation for the complete removal of the tetracycline resistance genes could be the removal and/or inactivation of the host bacteria. The genes were widely reported to be harbored by Bacteroides in the intestinal tract of humans and animals [28,31]. However, since the algal treatment system was highly aerated through both oxygenic photosynthesis and direct contact with the atmosphere, the obligate anaerobic bacterium would be eliminated [28].

#### 3.3.6. Macrolide-Lincosamide-Streptogramin Resistance Gene: *ermB*

In agreement with the findings of the current study, previous studies from wastewaters reported low abundance of this ARG in the influent [42], although more recent studies proved otherwise [11,43]. Similar to tetracycline resistance genes, *ermB* suffered early elimination from the system. Albeit not as much, the vulnerability of the gene to biological treatments was also reported by Nõlvak et al. (2018) and Hayward et al. (2019) in an algae-based system (2 LRV) and Lateral flow sand filters (5.4 LRV), respectively. Similar to tetracycline resistance genes, the actual force behind the early elimination of the gene remains unknown. A likely explanation would be the removal of bacterial perpetrators, Gram positive bacteria [44], of the gene. In order to unravel this, metagenomic analysis of the influents and effluents should be considered in future studies. 

#### 3.3.7. The Effect of Treatment on int1 Relative Abundance

The *int1* gene (class 1 integrase-encoding gene) was reported to be abundant in wastewaters due to its widespread distribution in humans and domestic animals. As a mobile genetic element present in both pathogenic and commensal bacteria (especially Gram negative) of human and animal origin [23,45], the gene serves as a biomarker for both the horizontal gene transfer of resistance genes and anthropogenic pollution [45,46,47]. As reported with past studies [11,48], the relative abundance of the *int1* gene in the influent stood higher than any other gene in the present study. This relatively high abundance of the gene in the system points to possible high turnover of gene exchange in the system. All the algal cultures (except KL10) witnessed enrichment (CK) to negligible reduction of the gene (CM2 and WW) in the first 4 days of treatment, which not only corroborates past findings [11,49] but also stresses the need for longer holding time for the appreciable reduction of the gene. This behavior of the gene in the first four days of treatment, which coincided the total elimination of *ermB* and tetracycline resistance genes in the system, suggests that *int1* could not have been related to the dissemination of those ARGs in the system. In most of the cultures, the reduction of the gene improved with time, showing the promise of the system in dealing with the gene. Beside *ermB* and the tetracycline resistance genes, which went undetected at day 4, *int1* was the most reduced gene in the entire system at the end of the treatment. This shows the ability of the system to effectively mitigate gene transfer among the host bacteria.

#### 3.3.8. The Effect of the Treatment on *uidA* Relative Abundance

The *uidA* gene, which codes for β-glucuronidase activity, has been considered as a surrogate marker for *E. coli* [47,50]. To date, *E. coli* has remained the most generally accepted proxy for potential pathogen dissemination in aquatic systems [50]. Evidently, all the cultures were found effective in reducing the gene right from the first four days of treatment. Importantly, the gene showed no sign of resurgence in the system with the passage of time and even the initial 4-day treatment would have sufficed for KL10 and the multi-species cultures. The highest LRV (2.364) favorably compared with 98.70% reduction of the gene reported for wastewater treatment plant [47]. Furthermore, in an algae-based system, persistence of *E. coli* was reported in a secondarily-treated wastewater utilizing *Nannochloropsis salina* (CCMP 1776) in an enclosed photobioreactor [50]. Despite this sterling performance, the *uidA* level as a surrogate marker for *E. coli* level should be treated with caution. This is because while the conventional culture technique for *E. coli* enumeration only considers culturable *E. coli* cells to be viable the PCR technique accounts for the genes in both viable and dead cells as well as in extracellular genetic materials [50].

### 3.4. Comparison of Genes Reduction Performance between Cultures

The individual cultures exhibited varied performances in the reduction of the relative abundance of the study genes. For *ermB*, *tetO*, *tetW* and *tetX*, which were eliminated by day 4 in all the cultures, it would be fair to conclude that all the cultures were equally promising. However, for the rest of the genes, comparison of the magnitude of the reduction at the end of the treatment revealed differences among the cultures. Firstly, the monocultures proved superior to the multi-species cultures in reducing the relative abundance of all the genes. However, this is with the exception of *floR* and *int1*, whose reduction in the best multi-species culture (WW) equaled that of the two monocultures. It is however not clear why the monocultures outperformed the multi-species cultures in terms of genes reduction. Evidently, the latter’s comparable performance in reducing the physicochemical parameters of the wastewater as well as biomass (TSS) accumulation testifies that the differences could not have arisen from performances in any of those parameters. In the screening step of this study, the monocultures were found to contain a diverse array of fatty acids as well as rich composition of carotenoids pigment (in KL10 monoculture). Studies have documented that monocultures of *Desmodesmus* sp. (same genus as KL10) produce bioactive compounds, such as carotenoids and fatty acids that are potent against *Staphylococcus aureus, Listeria monocytogenes,* methicillin resistant *S. aureus*, *Pseudomonas aeruginosa, Aeromonas hydrophila, E. coli, Bacillus subtilis, B. cereus* and *Streptococcus pyogenes* [51]. Similarly, a strain of *Coelastrella* sp. (same genus as CM2) with a fatty acids profile similar to that of CM2 strain has proved potent against *E. coli* UPEC, *P. aeruginosa* and *Klebsiella pneumoniae* [52]. However, no substantive conclusion can be drawn based on this, as the fatty acid composition of the multi-species cultures was not analyzed at the screening step of this study.

### 3.5. Relationships between ARGs, Indicator Genes and Water Quality Parameters

#### 3.5.1. Relationships between ARGs and Indicator Genes

The PCA was carried out in order to investigate the possible relationships between the study variables. As presented in the biplots, the study genes (except *floR*) clustered together, hinting at possible linkages between them, i.e., some might have been borne by the same mobile genetic elements and/or affected by the same biotic and abiotic conditions in the treatment system. It is be expected that ARGs belonging to the same family will be most closely related. However, in the current study, this was most true for *bla*_CTX_ and *bla*_TEM_*,* which consistently clustered together in all the biplots. Similar findings were reported previously [30,40] and it was revealed that the two genes are borne on the same plasmids [30]. This suggests that change in the concentration of one of the two genes in the system may help in predicting that of the other. Contrary to this, the relationship between the two sulfonamide resistance genes, *sul1* and *sul2*, was less consistent, ranging from insignificant to highly significant. This is consistent with previous studies [53,54]. It has also been observed that the two genes rarely co-occur in the same bacteria and are usually disseminated by different genetic materials [55,56,57]. Nevertheless, the relationship between *sul1* and *sul2* appeared rather firm compared with that between *cmlA* and *floR*. Although more closely related with each other than with any other gene, the two chloramphenicol resistance genes did not present any significant positive (*p* > 0.05) correlation with each other. These findings are concordant with previous ones [46,54,58] and would support the conclusion that the bacterial hosts of the two genes might have differed significantly.

Apart from the relationships between genes of the same family, there were also notable relationships between genes belonging to different families. The most prominent of these is the positive correlation between *uidA* and all other studied genes, indicating that the gene has the potential to serve as a biomarker for the prediction of ARGs pollution in the system. Better still, the gene expressed the most solid, consistent positive correlation with *bla*_CTX_, *bla*_TEM_ and *cmlA* and *aadA.* Previous studies have also reported the correlation of the two β-lactam with the plate counts of *E. coli*, prompting conclusion that the bacterium is the likely host of the two genes [30,40]. As for *cmlA* and *aadA*, their occurrence in *E. coli* strains has also been documented [59,60,61]. Altogether, the current study has therefore expanded on previous findings by demonstrating that the genetic marker of *E. coli* (*uidA*) could also serve as a reliable proxy for *bla*_CTX_, *bla*_TEM_*, cmlA and aadA* pollution in the system.

Besides *uidA*, the gene *int1* also expressed significant positive correlations with *sul1* and *aadA.* The *sul1* gene has been reported to be a component of *int1* cassette [11,32,46] and the relationship between the two has been well established [23,34,46]. Similarly, the observed strong relationship between *int1* and *aadA* is consistent with a number of previous findings [11,62,63].

#### 3.5.2. Relationships between ARGs and Water Quality Parameters

The study genes also revealed some correlation with the water quality parameters. Among these, the correlation of the study genes with TDS and EC was noteworthy. However, since TDS and EC are highly interrelated, discussing the results with regards to either of the two would suffice for both. TDS provides a measure of inorganic and organic solids, including minerals, salts, metals and ions, in a water medium. In the current study, TDS expressed positive correlations with all the study genes excluding *floR*. However, the firmest and most consistent of these relationships was observed with *sul1* and this agrees with previous findings [64,65]. This firm linkage of TDS with *sul1* suggests that TDS might be considered an indirect physicochemical indicator of *sul1* pollution in the treatment system. It has been reported that in surface waters, the abundance of bacterial genera, ARGs and TDS were positively correlated [66]. This suggests that in the current study the uptake of dissolved ions by microalgae might have made them less available for bacteria thereby leading to their reduced relative abundance of the study genes in the system.

In this study, contrary to TDS, TSS and pH rather correlated (negatively) with the study genes. TSS is a measure of microalgal biomass/growth in the system and its concentration expressed negative correlation with the relative abundance of all the study ARGs except *floR*. However, the strongest of these correlations were observed with *bla*_CTX_ (*p* < 0.01)*, bla*_TEM_ (*p* < 0.01), and *aadA* (*p* < 0.01). In a similar study, the concentration of chlorophyll a (also used as an index of algal growth) correlated negatively with the abundances of certain genes including *aadA* (*p* < 0.05), *int1* (*p* < 0.05), *sul1* (*p* < 0.05), *ermB* (*p* < 0.01) and *tetW* (*p* < 0.01). Interestingly, the strong negative correlation of chlorophyll a with *ErmB* and *tetW* might help explain the basis for the early disappearance (within 4 days) of the same ARGs, and probably *tetO* and *tetX*, in the current study. This is because in the present study, the first four days of treatment, which witnessed the elimination of the four genes, coincided with the period that recorded the fastest algal growth in the system (Figure 1A). However, further studies would be needed in order to establish the veracity of this assertion. The precise mechanism through which microalgal growth contributed to gene reduction is not clear; however, possible mechanisms may include (1) the adsorption of the bacterial cells to the microalgal biomass in the system [24]; (2) the gravity-induced settlement of microalgal-bacterial floccules, mediated by bacterial extracellular polymeric substances [24,67]; (3) the synergistic effects of dissolved oxygen and sunlight, leading to the formation of reactive oxygen species that cause bacterial cell damage [10]; and (4) the release of antibacterial metabolites by the microalgae [10].

Growth of most microalgae has been observed to lead to change in pH of the medium as a result CO_2_ uptake, thus yielding a firm positive correlation between pH and a specific growth parameter [68,69]. In the current study, the negative correlation observed between the study genes and pH implied that the increase in pH observed from day 0 until day 12 (for most cultures) played a key role in the reduction of ARGs. Similar strong negative correlation was observed between pH and genes, including *int1, sul1, aadA, ermB* and *tetW* (r = −0.91 to −0.97, *p* < 0.01 or *p* < 0.05). The reduction was attributed to a microbial community shift imposed by elevated pH (10.6 at day 16) resulting from accelerated microalgal growth [11]. Besides the shift in microbial community, the reduction of heterotrophic and pathogenic bacteria, that form the bulk of antibiotic resistant bacteria, at high pH, is another possibility [70]. It was observed that, in microalgae-based systems, pH in the range of 10 to 10.5 brought about appreciable removal of *E. coli*, fecal coliforms, enterococci and heterotrophic bacteria [18,70,71]. In this study, the maximum pH attained by all the microalgae cultures crossed the 10.5 mark and therefore it would be reasonable to conclude that, among other factors, pH might have contributed to the overall reduction in the relative abundance of the study genes. The precise mechanism of pH-mediated disinfection involves the entrapment of photon in the algal thylakoid membrane leading to the introduction of hydrogen ion while leaving out the corresponding hydroxyl ion [71]. This consistently raises the pH of the medium to high alkaline levels resulting in damage to bacterial cells [18,71].

## 4. Materials and Methods

### 4.1. Outdoor Cultivation of Microalgae Combinations in MWW

Raw MWW samples (influent) were collected in clean 25 L plastic containers at three different points along the sewer. These were immediately transported to the lab for analyses. The sampling site was a sewer adjacent to Ayub Teaching hospital, Abbottabad, Pakistan. The sewer passed through the hospital and the samples were collected at the point where the sewer just emerged from the hospital. The sample containers were then emptied into a large plastic vessel and mixed together. Finally, the samples were distributed in 2.8 L volumes into twelve 9.5 L plastic basins. The microalgae strains used in this study, included CM2 (*Coelastrella* sp. MW962206) and KL10 (*Desmodesmus* sp. OP279595) and a native, uncharacterized, wastewater microalgae consortium (WW) for comparison. Detailed genetic and morphologic characterization of the strains have been reported previously [72]. BG11 cultures of these were prepared and incubated for two weeks. Afterwards, inocula from these cultures were prepared and adjusted to the density of OD750 = 0.25. These were introduced into the designated wastewater containers in the ratio of 1: 15 (200 mL in 2800 mL) and mixed thoroughly. All the cultures were prepared in triplicate and kept in the open for 3 weeks. The cultures were mixed daily throughout the wastewater treatment period.

### 4.2. Determination of Physicochemical Water Quality Parameters 

The influent sample was analyzed for physicochemical parameters. The analyses of the pH, temperature, total dissolved solids (TDS) and electrical conductivity (EC) was conducted using a Combo waterproof tester. Total suspended solids (TSS) was analyzed using a sample passed through a filter glass fiber paper (0.2 μm) and oven-drying [73], while NO_3_-N was analyzed according to Ultraviolet Spectrophotometric Screening Method [73]. The initial levels of the parameters in the influent were pH (7.71 ± 0.1); temperature (15.93 °C ± 0.06), TDS (614.7 mg/L ± 0.58), EC (1229 μS/m ± 1), TSS (400 mg/L ± 26.5) and NO_3_-N (0.64 ± 0.08 mg/L). At 4-day time interval, sample effluents from the algal cultures were collected and analyzed for all the above-mentioned parameters.

### 4.3. Antibiotic Resistance Genes Pollution Studies

#### 4.3.1. Genomic DNA Extraction

This was carried out according to the modified protocol of Inuwa et al. (2022). Briefly, 45 mL of the sample (influent or effluent, as the case might be) was centrifuged (1 × 10^3^ rpm/20 min) and the supernatant was discarded. This was followed by the addition of sterile distilled water (500 µL) and 10% sodium dodecyl sulphate (400 µL). To this mixture, 80 µL of proteinase K was added, vortex-mixed briefly and incubated (56 °C /4 h) with regular shaking. This was followed by the addition of CTAB (320 µL) solution, containing 2% (w/v) cetyltrimethylammonium bromide (CTAB), 1.4 M NaCl, 100 mM Tris-HCl and 20 mM EDTA mixture (pH 8.0). After a brief vortex-mixing, NaCl solution (400 µL) was added and mixed again. Equal volume of phenol: chloroform: isoamyl alcohol (25:24:1) mixture was added, vortex-mixed and centrifuged (6 × 10^3^ rpm/10 min). The upper layer was collected and treated with phenol: chloroform: isoamyl alcohol mixture again. After centrifugation, the upper layer was again collected and equal volume of isopropanol was added and allowed for precipitation in a refrigerator (4 °C/overnight). Following centrifugation (13 × 10^3^ rpm/15 min), the supernatant was gently discarded. The pellet was washed with 70% ethanol (200 µL) and centrifuged as above. This step was repeated again and the pellet was dried in a Speedvac. The DNA was resuspended in a TE buffer, allowed to dissolve in a refrigerator for 24 hours. The integrity of the DNA extracts was confirmed using gel electrophoresis (2% agarose gel, 1× TAE buffer and 120 V/30 min). Finally, the samples were frozen (−20 °C) until needed.

#### 4.3.2. Qualitative Polymerase Chain Reaction

Prior to analysis, the concentration of all the DNA samples was adjusted to 40 ng/µL using a nanodrop spectrophotometer (Colibri, Titertek Berthold, Oak Ridge, TN, USA). The final reaction volume for each gene was 20 µL, consisting of master mix (10 µL), forward primer (0.8 µL), reverse primer (0.8 µL), DNA template (1 µL) and molecular grade water (7.4 µL). The initial qualitative detection of the genes of interest in the influent stream was carried out in a Thermocycler for each primer pair. The PCR conditions consisted of initial denaturation at 94 °C/5 min, denaturation 94 °C/30 s, annealing/30 s, extension at 72 °C/45 s and final extension at 72 °C/8 min. Details of the primer pairs’ sequences, annealing temperature and the number of cycles for each primer pair can be found in Appendix A.

#### 4.3.3. Real-Time Polymerase Chain Reaction

This was conducted on PIKOREAL 96 Real-Time PCR System (Thermo Scientific, Waltham, MA, US). The reaction volume was (20 µL) consisted of 10 µL qPCR Maxima SYBR green/ROX master mix (2×); forward primer (0.5 µL), reverse primer (1 µL) and water (8 µL). The cycling conditions for each primer pair was consisted of initial denaturation (94 °C/5 min); and 40 cycles of denaturation (95 °C/30 s), annealing (at a specified temperature for each primer pair) and 72 °C/8 min. To ascertain the specificity of the products, melt curve analysis was performed in the range of 55 to 95 °C. Each reaction was run in triplicates and negative control reaction (molecular grade water in place of DNA template) for each primer pair was also performed. The primer pairs sequence, annealing temperature and the number of cycles for each primer pair remain the same as in qualitative PCR.

For each microalgae culture, the gene copy number was calculated according to the equation, gene copy number = 10^(31−Cq)/(10/3)^, where Cq represents the threshold cycle [74]. The gene copy number of each target gene was then normalized by dividing with the corresponding 16S *rRNA* gene copy number [75]. The total genes’ relative abundance for each holding time period was calculated by summing up the average gene relative abundances of the individual microalgae cultures. Similarly, the LRV of the individual genes was derived from the relation log_10_ (relative abundance influent ÷ relative abundance effluent), where effluent represented the sample collected at the particular holding time [76].

### 4.4. Statistical Analyses

The means of the physicochemical parameters, nutrients and organic matter removals at day 20 were compared with one another using one-way ANOVA. Similarly, the means of gene relative abundances between days were compared using one-way ANOVA. Additionally, for each gene, the relative gene abundances of the individual cultures at a particular holding time were compared with one another using one-way ANOVA. In addition, comparison of gene abundances between any two relative abundances of genes at different holding times for a particular culture was achieved using a two-tailed paired samples *t*-test at 5% level of significance. Finally, all statistical analyses were conducted at 5% level of significance on SPSS version 25, while figures were plotted on Origin 2021.

## 5. Conclusions, Future Prospects, Limitations and Recommendations

### 5.1. Conclusions

Based on the findings of this study, the following conclusions can be drawn:Two indicator genes, *int1* and *uidA*, analyzed in this study proved reliable biomarkers for ARGs pollution in the microalgae-based WWT system. Interestingly, the *uidA* level could be used to monitor potential fecal and/or ARGs pollution. However, the relationship is most precise with *aadA*, *bla*_CTX_, *bla*_TEM_ and *cmlA*. In the same vein, *int1* proved to be a reliable biomarker for *sul1* and *aadA* genes.The abundance of ARGs in the system can be represented indirectly by the concentrations of TDS, EC and TSS. More specifically, TDS and EC related most with *sul1*, whereas TSS showed the most solid relationship with *bla*_CTX_, *bla*_TEM_ and *aadA*.Tetracycline resistance genes (*tetO, tetW* and *tetX*) and *ermB* were highly vulnerable to reduction in the microalgae-based WWT system, whereas *floR* was the most persistent.Although all the cultures proved promising in removing the conventional MWW quality parameters, the monocultures were the most efficient in terms of gene reduction from the MWW stream. This proves the superiority of the monocultures over the multi-species cultures, synthetic or natural, both in binary or polyculture combinations.

### 5.2. Future Prospects, Limitations and Recommendations

The present study has successfully established a linkage between the reduction of specific antibiotic resistance genes, *E. coli* (a bacterial indicator of water quality) and certain physicochemical water quality parameters. These findings consolidate the existing knowledge in the field and further contribute towards the evolution of the AWWT system as a green alternative that achieves the requirement of emerging contaminants (antibiotic resistance genes) removal from MWW.

Despite the success recorded, this study was not without a few limitations. Firstly, the removal of genes in the system was presumed to be principally achieved through the shift in the bacterial community under the influence of the biotic and abiotic forces in the system. However, this assertion could have been solidified through the tracking of the bacterial communities along the treatment line. That would have revealed the actual fate of the bacterial communities and allowed for a more concrete conclusion. Secondly, the rapid elimination of *tetO*, *tetW* and *tetX* and *ermB* genes by the system implied that the 4-day effluent analysis approach adopted herein appeared to be lengthy to allow periodic monitoring of the fate of those genes in the system. To circumvent the above limitations, future studies should include metagenomic analysis of both influents and effluents to reveal the actual fate of the bacterial community and therefore shed more light on the possible gene reduction mechanism(s) in the system. Furthermore, the length of the time lag between successive effluent analyses of all the study parameters should be lowered to fully accommodate any gene that might have the propensity for rapid disappearance from the system. Finally, to consolidate on the gains of the outcome of the current study, up-scaling on the current capacity should be considered in future efforts. This would give better reproducibility prospects of the outcome of the study when the system is put into real practice.

## Figures and Tables

**Figure 1 antibiotics-11-01531-f001:**
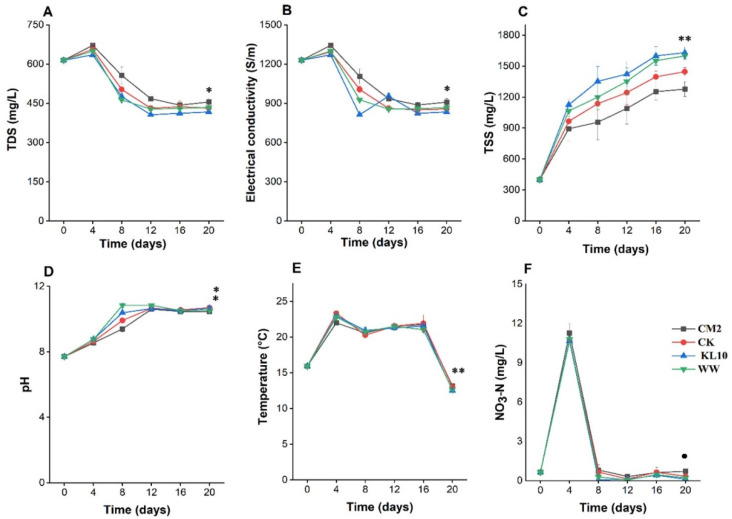
Time course changes in physicochemical parameters of the MWW due to microalgae growth (**A**–**F**). Error bars indicate standard deviation of triplicate readings for each culture treatment. TDS: total dissolved solids; TSS: total suspended solids; * indicates that the best monoculture was not more efficient (*p* > 0.05) than the best multi-species culture; ** indicates that the best monoculture was equal to the best multi-species culture; ⁑ indicates that the best multi-species culture (CK) was not more efficient than the best monoculture (KL10); • indicates that the best monoculture (KL10) was more efficient (*p* < 0.05) than all.

**Figure 2 antibiotics-11-01531-f002:**
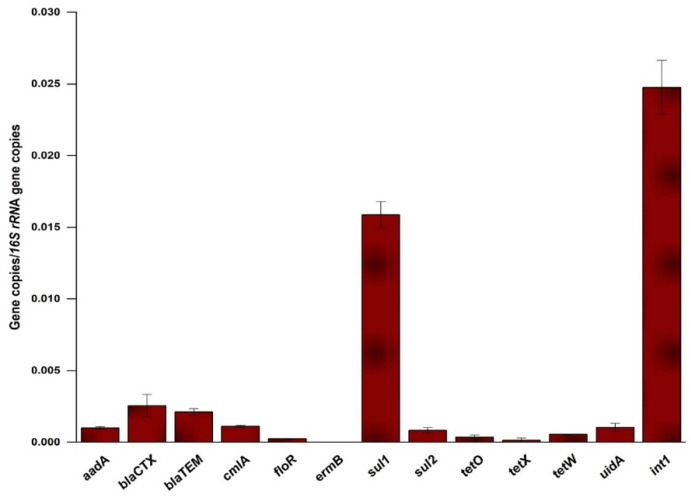
Relative abundance of the individual genes evaluated in the MWW effluent. Error bars indicate the standard deviation of replicated readings.

**Figure 3 antibiotics-11-01531-f003:**
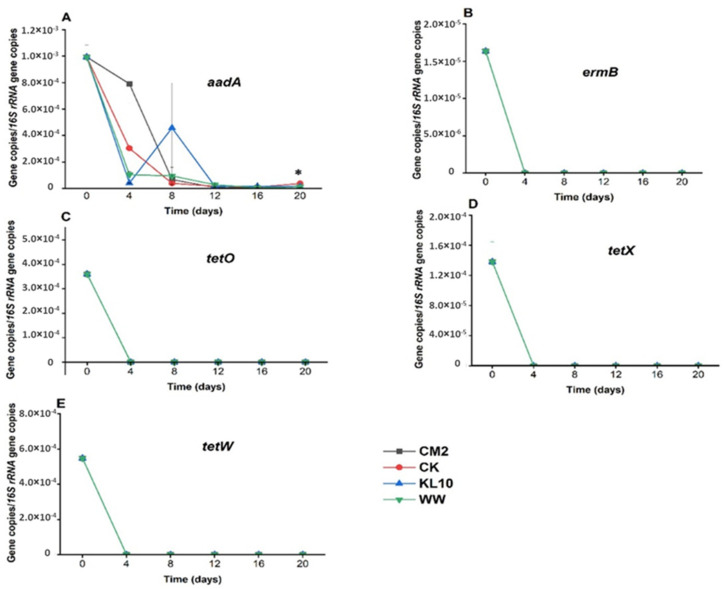
Time course changes in relative abundance of *aadA* (**A**), *ermB* (**B**) and representative tetracycline resistance genes (**C**–**E**) in the wastewater due to microalgae growth. Error bars indicate standard deviation of triplicate readings for each culture treatment. * indicates that one of the monocultures (CM2) performed better than all others (*p* < 0.05).

**Figure 4 antibiotics-11-01531-f004:**
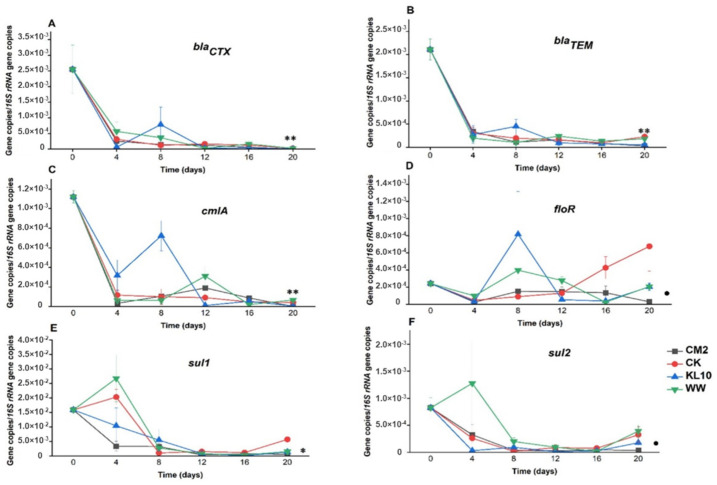
Time course changes in relative abundance of representative β-lactam (**A**,**B**), phenicols (**C**,**D**) and sulfonamide (**E**,**F**) resistance genes in the wastewater due to microalgae growth. Error bars indicate standard deviation of triplicate readings for each culture treatment. * indicates that monoculture (CM2) performed better than all others (*p* < 0.05). ** indicates that monocultures performed better than multi-species cultures (*p* < 0.05). • indicates that monocultures were not more efficient (*p* < 0.05).

**Figure 5 antibiotics-11-01531-f005:**
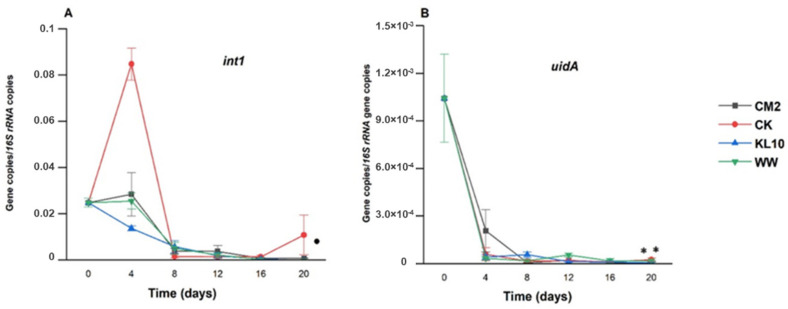
Time course changes in relative abundance of *int1* (**A**) and *uidA* (**B**) in the wastewater due to microalgae growth. Error bars indicate standard deviation of triplicate readings for each culture treatment. • indicates that the best monoculture did not perform better than the best multi-species culture (*p* > 0.05). ** indicates that the two monocultures performed better than the two multi-species cultures (*p* < 0.05).

**Figure 6 antibiotics-11-01531-f006:**
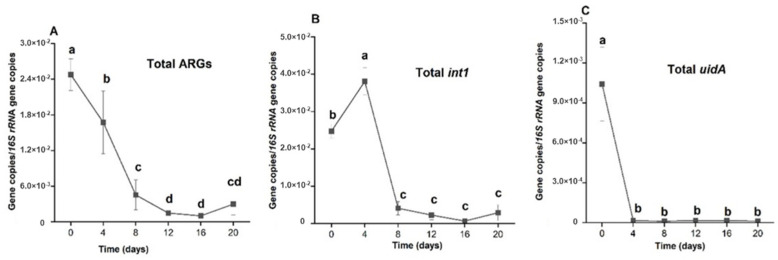
Comparison of the cumulative gene reduction efficiencies for total ARGs (**A**), total *int1* (**B**) and total *uidA* (**C**) at different holding time. Error bars indicate standard deviation of triplicate readings for each holding time. Different letters indicate significant difference (*p* > 0.05) in the total relative abundances of the genes between the holding times.

**Figure 7 antibiotics-11-01531-f007:**
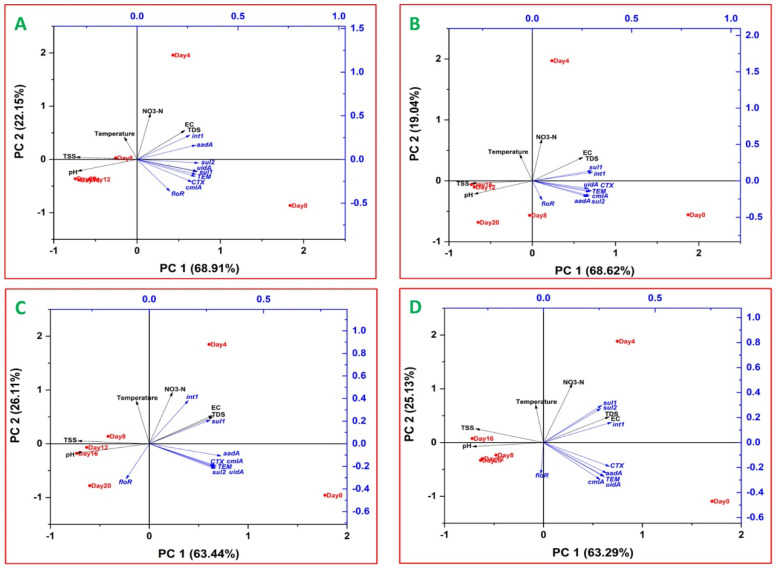
Principal component analyses of relative abundances of ARGs, *int1* and *uidA* (gene copies*/*16S*rRNA)* and water quality indicators at different times in CM2 (**A**), KL10 (**B**), CK (**C**) and WW (**D**) cultures.

## Data Availability

The data presented in this study, was handled by SPSS v25 and Origin 2021 for graphical representation while the raw data are available on request from the corresponding author.

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
