# Peer review of "Removal of Antibiotic Resistance Genes, Class 1 Integrase Gene and Escherichia coli Indicator Gene in a Microalgae-Based Wastewater Treatment System"

_antibiotics, 2022, doi:10.3390/antibiotics11111531_

Round 1

Reviewer 1 Report

This paper looks at the removal of Antibiotic resistance genes rom wastewater by treatment with microalgae. The paper overall contains some interesting results, minor changes are required before publication.

Major Changes

Please give more information on the mircoalgae- What species are they ? Where were they isolated from?, etc.

Minor Changes

Line 58 Replace "These potentials for ARGs" with "This potential for ARG"

Might be good to add a small explanation of what the Microaglae are early in the text 

In Figure 1 add a key to explain TDS, EC and TSS.

Line 124 - "The temperature"

Line 336 - Beta- Lactamases are not always plasmid based.

Author Response

Reviewer-1

This paper looks at the removal of Antibiotic resistance genes from wastewater by treatment with microalgae. The paper overall contains some interesting results, minor changes are required before publication.

Major Changes

Please give more information on the mircoalgae- What species are they? Where were they isolated from? etc.

Response: This has been appropriately responded (please check MM section particularly lines 645 to 646 of the revised manuscript)

Minor Changes

Line 58 Replace "These potentials for ARGs" with "This potential for ARG"

Response: This has been addressed (please refer to line 60-61 of the revised manuscript).

Might be good to add a small explanation of what the Microaglae are early in the text

Response: The role of microalgae in wastewater remediation has been added in order to supplement the existing report on ARGs removal (please check lines 63 to 69 of the revised manuscript)

In Figure 1 add a key to explain TDS, EC and TSS.

Response: This has been added (please refer to line 93 of the revised manuscript).

Line 124 - "The temperature"

Response: This has been corrected (please check line 138 of the revised manuscript).

Line 336 - Beta- Lactamases are not always plasmid based.

Response: This has been dealt with appropriately.

Reviewer 2 Report

The study describes the "Removal of Antibiotic Resistance Genes, Class 1 Integrase Gene and Escherichia coli Indicator Gene in Microalgae-based Wastewater Treatment System"

The study is very lengthy while saying very little that is specific especially the discussion part. I would suggest condensing the discussion part. The manuscript should make some recommendations.

1.  Add some related methodology in the introduction section. Take help and cite the following relevant studies in introductory part. 

  • DOI: 10.1016/j.chemosphere.2019.124680

  • DOI: 10.1016/j.envpol.2021.116603

doi: 10.1007/s11356-022-22197-4

2.  Line 124. "he" corrects the word to "The"

3. Line 138-139:  "The PCR assay targeted and detected a total of 11 ARGs, int1 gene and Escherichia coli gene, uidA, in the influent."  Mention those ARGs which were detected. Why only E. coli gene ? Why not other bacteria? uidA is a Universal target gene for E. coli detection. Also rephrase the sentence.

4. Line 139-141: If microalgae cultivation in the municipal water have no effect on the the genes quantified both in influents and effluents then what is the benefit of this study ?

5. Why only few genes were selected for detection in this study? As there are number of mobile elements and plasmids involved. What was the criteria ?

6. Line 163: , blaCTX, blaTEM, correct the format of gene name throughout the manuscript. bla will be italicized while CTX and TEM should not be italicized

7. Line 205: "int1and" give space between two words

8. Discussion part is very lengthy. It is not easy to read it in the current form. I would suggest to make it short. Explain and compare only the important results you found otherwise for readers there will be no interest to read. Please revise it carefully.

9. Material methods: Line 581: How many samples were collected from wastewater?

10. What was the temperature you stored the wastewater samples for 3 weeks?

11.  Line 592. 4.2. Determination of Physicochemical Water Quality Parameters: I did not found the physiochemical properties of wastewater in the results section. Please check and revise.

12. Line 603: 4.3.1. Genomic DNA Extraction: Insert the references in this section

Author Response

Reviewer-2

The study describes the "Removal of Antibiotic Resistance Genes, Class 1 Integrase Gene and Escherichia coli Indicator Gene in Microalgae-based Wastewater Treatment System"

The study is very lengthy while saying very little that is specific especially the discussion part. I would suggest condensing the discussion part. The manuscript should make some recommendations.

Response: This has been dealt appropriately to shorten and comprehend the discussion, please see the revised manuscript.

  1. Add some related methodology in the introduction section. Take help and cite the following relevant studies in introductory part.  

DOI: 10.1016/j.chemosphere.2019.124680

DOI: 10.1016/j.envpol.2021.116603

doi: 10.1007/s11356-022-22197-4

Response: Thanks for the suggestion, these papers focused on ARGs, have been cited appropriately.

  1. Line 124. "he" corrects the word to "The"

Response: This has been corrected (please check line 138 of the revised manuscript)

  1. Line 138-139:  "The PCR assay targeted and detected a total of 11 ARGs, int1 gene and Escherichia coli gene, uidA, in the influent." Mention those ARGs which were detected. Why only E. coli gene? Why not other bacteria? uidA is a Universal target gene for E. coli detection. Also rephrase the sentence.

Response: The studied ARGs have been listed (please refer to line 152 to 153 of the revised manuscript). As for the choice of E. coli gene, this bacterium serves as a universal biomarker of probable fecal pollution and is routinely monitored in wastewater treatment plants. The purpose herein was to link it with ARGs within the same system.

  1. Line 139-141: If microalgae cultivation in the municipal water have no effect on the genes quantified both in influents and effluents then what is the benefit of this study?

Response: Temporal quantification of the studied genes and other parameters was carried out in the current work. And, the effect of algal growth in wastewater stream was obvious on the relative abundance of tested genes. We hope the confusion caused is better settled now in revised ms.

  1. Why only few genes were selected for detection in this study? As there are number of mobile elements and plasmids involved. What was the criteria?

Response: As there are hundreds of bacterial resistance genes, our study picked some representative ones that are of global interest and we highlighted this earlier in the introduction section (e.g. see Hendriksen et al. 2019).

  1. Line 163: blaCTX, blaTEM, correct the format of gene name throughout the manuscript. blawill be italicized while CTX and TEM should not be italicized

Response: This has been corrected throughout (e.g. please refer to the revised manuscript lines 274-282 of the revised manuscript)

  1. Line 205: "int1and" give space between two words

Response: This has been corrected.

  1. Discussion part is very lengthy. It is not easy to read it in the current form. I would suggest to make it short. Explain and compare only the important results you found otherwise for readers there will be no interest to read. Please revise it carefully.

Response: The discussion has been shortened and comprehended now, please see the revised manuscript.

  1. Material methods: Line 581: How many samples were collected from wastewater?

Response: This has been answered (please refer to page 638-644 of the revised manuscript)

  1. What was the temperature you stored the wastewater samples for 3 weeks?

Response: Samples were not stored, but processed immediately after collection. The temperature fluctuations along the treatment line has been presented in Fig. 1E, of the revised manuscript.

  1. Line 592. 4.2. Determination of Physicochemical Water Quality Parameters: I did not found the physiochemical properties of wastewater in the results section. Please check and revise.

Response: Total suspended solids, electric conductivity, pH and temperature are the physicochemical parameters analyzed in the current study. Similar referrals may be found in the works listed below:

https://doi.org/10.1016/j.jhazmat.2020.123795

https://doi.org/10.1016/j.jhazmat.2019.121221

https://doi.org/10.1016/j.jes.2018.09.004

  1. Line 603: 4.3.1. Genomic DNA Extraction: Insert the references in this section

Response: This has been provided. (Please refer to line 669 of the revised manuscript)

Reviewer 3 Report

The authors wrote the manuscript very well. Here are some comments:

1. Give the full name before the abbreviation. For example, TSS, TDS in the abstract.

2. Introduction: give more previous work on using Microalgae to remove antibiotic resistance genes. (However, authors compare the result with other work in discussion)

3. Method: The authors should explain more about using strains, KL10 and CMS. Are they algae? Which genus are they belong to?

4. Result and discussion are acceptable.

Author Response

Reviewer-3

The authors wrote the manuscript very well. Thanks for your encouragement. Here are some comments:

  1. Give the full name before the abbreviation. For example, TSS, TDS in the abstract.

Response: This has been addressed (see the revised manuscript).

  1. Introduction: give more previous work on using Microalgae to remove antibiotic resistance genes. (However, authors compare the result with other work in discussion)

Response: The relevant literature has been added in the introduction section as well (Refer to lines 64 to 68 of the revised manuscript).

  1. Method: The authors should explain more about using strains, KL10 and CMS. Are they algae? Which genus are they belong to?

Response: This has been appropriately responded; please see lines 644-646 of the revised manuscript.

  1. Result and discussion are acceptable.

Response: Thanks for your encouragement.

Reviewer 4 Report

The manuscript by Inuwa et al. titled "Removal of Antibiotic Resistance Genes, Class 1 Integrase Gene and Escherichia coli Indicator Gene in Microalgae-based Wastewater Treatment System" presents an analysis of reduction the relative copy number of ARGs, uidA and intI1 from municipal wastewater in a microalgae-based wastewater treatment system. The authors also conducted an analysis of the relationship of the studied genes with the physicochemical properties of the wastewater, during treatment. Although this research is not very innovative, it consolidates the current knowledge in this field. This manuscript has some shortcomings that the authors are aware of, but it is a coherent study. However, there are a few aspects that the authors need to pay attention to and improve. 

Major comments:

  • I have the impression that the authors did not pay attention to the details and overall drafting of the manuscript. In some places the manuscript was written using rather colloquial terms, especially in the "results" section. 

  • The manuscript should be revised in terms of language

  • in my opinion, chapter 3.3 should contain a coherent discussion of the obtained results. The authors often used to the same references when discussing the results of different ARGs. Combining these subsections would have a better effect and would make the results easier to analyze and read. 

  • the literature list is prepared very sloppily, double citations of the same works appear, e.g. "OsiÅ„ska et al."

Minor comments:

  • why were figures 3, 4 and 5 and 7 and 8 and also 9 and 10 separated. what did the authors want to achieve by this?

  • L169-171: misspelled sentence "was observed with floR reduction in cmlA", please correct

  • L248 - please add "r=" in parentheses

  • Fig. 5 and 6 why are there such long gaps between figure and description

  • in my opinion, fig. 9 and 10, which concern correlations, are not needed in the main text. In general, we can observe some correlation relationships in fig. 7 and 8 (PCA analyses), so fig. 9 and 10 seem to be a partial repetition. I would move these fig. to supplementary materials.

  • L303: please expand the abbreviation "NEQS" in the text

  • L303" "similary" to which this similarity refers, since in the previous sentence the results show a different character

  • L313 double space, 

  • throughout the manuscript there are places with double spaces, extra spaces or no spaces between words (e.g. 322, 328, 426,427,439,524, 464, 537, 545...)

  • L315: why in the first place the authors compare the results with those obtained from sewage in sweden? different climate zones, habits of people, nature of sewage....

  • L315, 595: why did the authors use different reference styles? please check throughout the manuscript

  • L319: please add "genes" after "all"

  • L328,329: shouldn't there be higher levels of gene copy number since the authors in the previous subsection assume that sul1, blaCTX and blaTEM genes are more due to "hospital waste-contamitated water"? this is a little inconsistent

  • L359-361" where are the results of NO3-N concentrations for the studied systems presented?

  • L378: why do the authors so emphasize the similarity of the results to trends from Asia and Africa, what about other continents? if there is a reason for this, please explain this in the manuscript

  • L381: "floR enjoyed" big colloquialism

  • subsections 3.3.4 and 3.3.6 why are the tested genes listed in the titles and not otherwise?

  • L468: references should be added to support this sentence

  • L519: combine references into one parenthesis

  • L542,543: there is a repeat information about the correlation between TSS and ARGs

  • L554-556: two points "1"

  • L563 please add "and" between "study genes" and "pH"

  • L571: "Enterococci" in small letter

Author Response

Reviewer-4

The manuscript by Inuwa et al. titled "Removal of Antibiotic Resistance Genes, Class 1 Integrase Gene and Escherichia coli Indicator Gene in Microalgae-based Wastewater Treatment System" presents an analysis of reducing the relative copy number of ARGs, uidA and intI1 from municipal wastewater in a microalgae-based wastewater treatment system. The authors also conducted an analysis of the relationship of the studied genes with the physicochemical properties of the wastewater, during treatment. Although this research is not very innovative, it consolidates the current knowledge in this field. This manuscript has some shortcomings that the authors are aware of, but it is a coherent study. However, there are a few aspects that the authors need to pay attention to and improve. 

Response: Thanks for your valuable feedback; we have improved the ms quality accordingly.

Major comments:

  • I have the impression that the authors did not pay attention to the details and overall drafting of the manuscript. In some places the manuscript was written using rather colloquial terms, especially in the "results" section. The manuscript should be revised in terms of language.

Response: Thanks for your suggestions; the revised ms has been improved linguistically, particularly, our coauthor from Canada (Dr. Emilie Widemann) has validated the revised ms draft for the betterment of English language.

  • In my opinion, chapter 3.3 should contain a coherent discussion of the obtained results. The authors often used to the same references when discussing the results of different ARGs. Combining these subsections would have a better effect and would make the results easier to analyze and read. 

Response: The section is revised as per comments to comprehend. But, we did not merged the subsections because the overall structure follow this way. Hopefully, the revised ms would read better than the previous.

  • The literature list is prepared very sloppily, double citations of the same works appear, e.g. "OsiÅ„ska et al."
  • Response: Sorry for this; rightfully the references OsiÅ„ska et al. and Wen et al. were mistakenly duplicated. These and other needful corrections have been keenly considered now. Please see the reference list of the revised ms.

Minor comments:

  • why were figures 3, 4 and 5 and 7 and 8 and also 9 and 10 separated. what did the authors want to achieve by this?

Response: To deal with this, the figures are now grouped based on certain similarity they share; for instance, in Fig.4, ARGs in the same row belong to the same family, while in Fig.5 the two figures are both for biomarker/indicator genes. While, the figures for correlation analyses are now included in ms as supplementary information.

  • L169-171: misspelled sentence "was observed with floR reduction in cmlA", please correct

Response: This has been addressed (please refer to line 186 of the revised manuscript).

  • L248 - please add "r=" in parentheses

Response: This has been done (please refer to the revised manuscript).

  • 5 and 6 why are there such long gaps between figure and description

Response: The excess gap has been trimmed now (please refer to the revised manuscript).

  • In my opinion, fig. 9 and 10, which concern correlations, are not needed in the main text. In general, we can observe some correlation relationships in fig. 7 and 8 (PCA analyses), so fig. 9 and 10 seem to be a partial repetition. I would move these fig. to supplementary materials.

Response: The figures in question have been moved to supplementary section (please refer to S1 and S2 figs in the supplementary documents).

  • L303: please expand the abbreviation "NEQS" in the text

Response: This has been treated (please refer to the revised manuscript).

  • L303" "similarly" to which this similarity refers, since in the previous sentence the results show a different character

Response: The word has been replaced with “however” (Please refer to line 315 of the revised manuscript).

  • L313 double space, throughout the manuscript there are places with double spaces, extra spaces or no spaces between words (e.g. 322, 328, 426,427,439,524, 464, 537, 545...)

Response: All the double spaces have been removed.

  • L315: why in the first place the authors compare the results with those obtained from sewage in Sweden? different climate zones, habits of people, nature of sewage....

Response: The main motive behind the comparison was to show that the range of ARGs relative abundance in the influent of the current study fell within the ranges reported by other related studies on ARGs while the geographical location and human habits were not of interest in this comparison.

  • L315, 595: why did the authors use different reference styles? please check throughout the manuscript
  • Response: The needful corrections have been considered now. Please see the reference list of the revised ms.
  • L319: please add "genes" after "all"
  • Response: This has been added (line 331 of the revised manuscript).
  • L328,329: shouldn't there be higher levels of gene copy number since the authors in the previous subsection assume that sul1, blaCTX and blaTEM genes are more due to "hospital waste-contaminated water"? this is a little inconsistent
  • Response: To add further clarity, the current study agrees with the previous ones on the high levels of sul1, blaCTX and blaTEM genes and low levels of tetracycline resistance genes.
  • L359-361" where are the results of NO3-N concentrations for the studied systems presented?
  • Response: This has been added, please see L142-150 of the revised ms.
  • L378: why do the authors so emphasize the similarity of the results to trends from Asia and Africa, what about other continents? if there is a reason for this, please explain this in the manuscript
  • Response: The regional comparisons are not the scope of this work. Sorry for any confusion earlier, we tried to deal this appropriately now (please refer to lines 171 to175 of the revised manuscript).
  • L381: "floR enjoyed" big colloquialism

Response: This has been corrected (please refer to line 410 of the revised manuscript).

  • Subsections 3.3.4 and 3.3.6 why are the tested genes listed in the titles and not otherwise?

Response: The genes hitherto unlisted have also been listed (please see the concerned subsections in the revised manuscript).

  • L468: references should be added to support this sentence

Response: This has been done (please see line 497 of the revised manuscript).

  • L519: combine references into one parenthesis

Response: This has been corrected (please refer to line 548 of the revised manuscript).

  • L542, 543: there is a repeat information about the correlation between TSS and ARGs

Response: This has been corrected (please refer to the revised manuscript).

  • L554-556: two points "1"

Response: This has been corrected (please refer to lines 698 to 711).

  • L563 please add "and" between "study genes" and "pH"

Response: This has been done (please refer to line 586 of the revised manuscript).

  • L571: "Enterococci" in small letter

Response: This has been corrected (please refer to line 652 of the revised manuscript).

Reviewer 5 Report

Thank you for the opportunity or review a manuscript entitled: "Removal of Antibiotic Resistance Genes, Class 1 Integrase 2 Gene and Escherichia coli Indicator Gene in Microalgae-based 3 Wastewater Treatment System".

The manuscript is of importance. The aim and the objectives are current and of worldwide interest. The writing of the manuscript is proper. The methodology is transparent. Minor remarks are presented below:

Introduction:

Introduction and the entire manuscript - please check the abbreviations - sometimes the authors use WW (wastewater) when they mean WWT - wastewater treatment, or WWTP (wastewater treatment plant) when they mean just wastewater. All abbreviations are valid and proper but sometimes used interchangeably where they shouldn't be. 

Please add one more paragraph to the introduction containing the main aim of the presented study - since, after the introduction, the manuscript goes straight to the results, it would be beneficial for future readers to see the study's goal before the results. 

Results: 

The presentation of the results is very clear. The formulation of conclusions is also proper. However, I would advise a slight extension of the discussion part - perhaps the authors should add some comparisons to other WWT technologies - by doing so, readers interested in such technique would see how it can compare to other methods of ARGs removal. 

Author Response

Reviewer-5

Thank you for the opportunity or review a manuscript entitled: "Removal of Antibiotic Resistance Genes, Class 1 Integrase 2 Gene and Escherichia coli Indicator Gene in Microalgae-based 3 Wastewater Treatment System".

The manuscript is of importance. The aim and the objectives are current and of worldwide interest. The writing of the manuscript is proper. The methodology is transparent.

Response: Thanks for your encouraging remarks.

Minor remarks are presented below:

Introduction:

Introduction and the entire manuscript - please check the abbreviations - sometimes the authors use WW (wastewater) when they mean WWT - wastewater treatment, or WWTP (wastewater treatment plant) when they mean just wastewater. All abbreviations are valid and proper but sometimes used interchangeably where they shouldn't be.

Response: This has been addressed (please refer to the revised manuscript). 

Please add one more paragraph to the introduction containing the main aim of the presented study - since, after the introduction, the manuscript goes straight to the results, it would be beneficial for future readers to see the study's goal before the results. 

Response: This has been added (please refer to lines 62-68 of the revised manuscript).

Results: 

The presentation of the results is very clear. The formulation of conclusions is also proper. However, I would advise a slight extension of the discussion part - perhaps the authors should add some comparisons to other WWT technologies - by doing so, readers interested in such technique would see how it can compare to other methods of ARGs removal. 

Response: Thank you for your comment, but to our understanding these comparisons already exist in the manuscript (please refer to lines 365 to 497 of the revised manuscript).

Round 2

Reviewer 2 Report

Thanks for addressing all the concerns and suggestions.

Author Response

Reviewer-2

Thanks for addressing all the concerns and suggestions.

Response: We really appreciate that the respected reviewer is satisfied with our responses.

Reviewer 4 Report

Thank you for addressing all the comments.

Nevertheless, I have a few more.

After revising the manuscript, the quality of all graphics dropped dramatically. The authors also do not stick to the article template. 

The authors added the results of NO3-N level concentration tests but did not add the method of its determination in the Materials and Methods section 

And one note for the future. When proofreading the manuscript, please mark in color only the applied changes. In this case, the authors marked the entire paragraph or subsections, making it difficult to review. 

Author Response

Reviewer-4

Thank you for addressing all the comments.

Response: We really appreciate that our responses are positively considered by respected reviewer.

Nevertheless, I have a few more. After revising the manuscript, the quality of all graphics dropped dramatically. The authors also do not stick to the article template. 

Response: We feel sorry for any “Graphics’ quality deterioration” that now has been improved with minimum 300dpi. We hope the current version of the manuscript is according to article template of MDPI-Antibiotics. Any further correction will surely be taken into consideration before publication of the final draft.

The authors added the results of NO3-N level concentration tests but did not add the method of its determination in the Materials and Methods section.

Response: The needful addition has been made in this version of the manuscript; please see Lines 681-684.

And one note for the future. When proofreading the manuscript, please mark in color only the applied changes. In this case, the authors marked the entire paragraph or subsections, making it difficult to review.

Response: We appreciate this suggestion. But, as a matter of fact, our manuscript was initially reviewed by five different reviewers and therefore the revision (v1) was quite extensive with significant modifications that we corresponded the marked paragraphs/subsections. v2 is not that extensive and so will hopefully be identifiable.